# Global syndromes induced by changes in solutes of the world's large rivers

Jiang Wu [1,2,8], Nan Xu [2,8], Yichu Wang [3], Wei Zhang [4], Alistair G. L. Borthwick [5,6] & Jinren Ni [1,7✉]

Solute-induced river syndromes have grown in intensity in recent years. Here we investigate seven such river syndromes (salinization, mineralization, desalinization, acidification, alkalization, hardening, and softening) associated with global trends in major solutes ($Ca^{2+}$, $Mg^{2+}$, $Na^+$, $K^+$, $SO_4^{2-}$, $Cl^-$, $HCO_3^-$) and dissolved silica in the world's large rivers (basin areas $\geq$ 1000 $km^2$). A comprehensive dataset from 600 gauge stations in 149 large rivers reveals nine binary patterns of co-varying trends in runoff and solute concentration. Solute-induced river syndromes are associated with remarkable increases in total dissolved solids (68%), chloride (81%), sodium (86%) and sulfate (142%) fluxes from rivers to oceans worldwide. The syndromes are most prevalent in temperate regions (30~50°N and 30~40°S based on the available data) where severe rock weathering and active human interferences such as urbanization and agricultural irrigation are concentrated. This study highlights the urgency to protect river health from extreme changes in solute contents.

[1] State Environmental Protection Key Laboratory of All Materials Fluxes in River Ecosystems, Peking University, 100871 Beijing, P. R. China. [2] The Key Laboratory for Heavy Metal Pollution Control and Reutilization, School of Environment and Energy, Peking University Shenzhen Graduate School, 518055 Shenzhen, P. R. China. [3] College of Water Sciences, Beijing Normal University, 100875 Beijing, P. R. China. [4] Department of Plant, Soil and Microbial Sciences, Michigan State University, East Lansing, MI 48824, USA. [5] Institute of Infrastructure and Environment, School of Engineering, The University of Edinburgh, The King's Buildings, Edinburgh EH9 3JL, UK. [6] School of Engineering, Computing and Mathematics, University of Plymouth, Drake Circus, Plymouth PL4 8AA, UK. [7] Center for Global Large Rivers, School of Environmental Science and Engineering, Southern University of Science and Technology, 518055 Shenzhen, P. R. China. [8] These authors contributed equally: Jiang Wu, Nan Xu. ✉email: jinrenni@pku.edu.cn

Solutes are among the most mobile and abundant materials in rivers and are vital to the health of freshwater ecosystems[1–3]. Conventionally, such solutes are dominated by calcium ($Ca^{2+}$), magnesium ($Mg^{2+}$), sodium ($Na^+$), potassium ($K^+$), sulfate ($SO_4^{2-}$), chloride ($Cl^-$), bicarbonate ($HCO_3^-$), and dissolved silica (DSi) ions. Estimates of the annual worldwide discharge of total dissolved solids (TDS) from rivers to oceans range from 3600 to 3800 million tons[2,4,5]. These solutes largely derive from mineral weathering by carbonic acid[6], and collectively account for over 95% of TDS in natural waters[7].

Previous studies of solutes and their changes are limited to either a few dissolved ions[8–10] or small spatial scales[3,11,12]. Gibbs[13] proposed that three major natural mechanisms control the chemistry of surface waters: atmospheric precipitation, rock weathering, and the evaporation–crystallization process. Edmond and Huh[14] suggested that bedrock lithology played a dominant role, and Berner and Kothavala[15] pointed out the importance of climate in chemical weathering. However, only about 17% of the present-day continental surface appeared free from direct human footprint[16]. Although acid deposition control measures reduced global freshwater $Ca^{2+}$ concentrations (even to levels approaching the threshold required by aquatic organisms)[8], other human activities raised $Na^+$, $K^+$, $SO_4^{2-}$, $HCO_3^-$, and $Cl^-$ concentrations in rivers[2,6,11,17–19]. Moreover, large dams have modified the solute content of rivers, as evidenced by the recent decreases in DSi in the Yangtze[20] and TDS in the Yellow River[21].

Changes in solute fluxes caused by large-scale negative interactions between humans and the environment often result in so-called "global syndromes" once certain thresholds are exceeded[22,23]. In the early 2000s, Meybeck[16] defined river syndromes as negative feedbacks of river systems to human activities, primarily including river fragmentation, flow regulation, water-sediment imbalance, and chemical and microbial contamination processes. Chen[7] investigated acidification, salinization, and alkalization in Chinese rivers, observing their association with coal consumption, agriculture, and hydraulic projects. Kopáček et al.[24] showed how regional socio-economic and land-use changes modified solute concentrations in the Vltava River, Czech Republic. More recently, Kaushal et al.[3] developed the concept of a freshwater salinization syndrome which links salinization and alkalinization processes along hydrologic flow paths from small watersheds to coastal waters. Using the long-term time series of specific conductance, pH, alkalinity, and base cation concentrations, Kaushal et al. investigated freshwater salinization syndromes in hundreds of stream and river sites throughout the continental United States[3]. Best[1] summarized typical anthropogenic stressors on the world's large rivers, which are of primary importance to varying solutes[8,25] and related river syndromes[26].

Global trends in material fluxes through rivers not only provide estimates of riverine material budgets and changing terrestrial/oceanic environments[9,27] in earth systems but also help to assess the status of river health[1,20,28] and identify anthropogenic drivers of such changes[11,29]. Based on a unique dataset of major solutes ($Ca^{2+}$, $Mg^{2+}$, $Na^+$, $K^+$, $SO_4^{2-}$, $Cl^-$, $HCO_3^-$) and dissolved silica derived from 600 gauge stations in 149 rivers (basin areas ≥1000 km²), we proposed a framework of notable solute-induced river syndromes (salinization, mineralization, desalinization, acidification, alkalization, hardening, and softening) which could be identified using thresholds of solute concentrations and associated trends in the world's large rivers. Our study provides a dynamic perspective by which to determine the occurrence of solute-induced river syndromes and how they develop with co-varying trends in runoff and solutes. This is of great significance in the maintenance of river ecosystem health.

## Results and discussion

**Co-varying trends in annual runoff ($Q$) and solute concentration ($C$) of the world's large rivers.** As the primary flux transported by rivers, the trend in $Q$ plays a crucial role in the evolution of solutes, e.g., through enrichment or dilution[3]. Figure 1 shows the co-varying trends of $Q$ and TDS concentration ($C_{TDS}$) in global rivers. Data for TDS in 149 world's large rivers (Fig. 1a) and trend analysis for $Q$ variation (Fig. 1b, statistically stable, decreasing, and increasing), revealed nine co-varying trends in $Q$ and TDS, and their patterns of occurrence in each of six continents (Fig. 1c). For the 379 hydrologic stations (Fig. 1c), 83% maintained a stable trend in $Q$, dominated by the [$Q$, $C_{TDS}$] co-varying patterns I (stable $C_{TDS}$ trend, 59%), IV (decreasing $C_{TDS}$ trend, 12%), and VII (increasing $C_{TDS}$ trend, 12%), which mostly occurred in Asia and Europe. At these stations, the variation of $C_{TDS}$ is primarily controlled by changes in solutes themselves. The remaining six [$Q$, $C_{TDS}$] co-varying patterns were observed at only 32 stations (8.4%) scattered in Asia and Oceania where the $C_{TDS}$ variation is the combined consequence of changes in both solutes and $Q$.

At decadal scale, the [$Q$, $C_{TDS}$] co-varying trends may experience strong shifts due to changes in river dynamics typically caused by anthropogenic stressors[1,16] and so a viable approach is required to identify large rivers suffering anthropogenic syndromes. For example, the increasing occurrence of Pattern VIII in Asia is associated with widespread alarming trends in river discharge reduction and $C_{TDS}$ increase and is especially prominent at stations in the middle and lower Yellow River in China. As suspended sediment concentration in the Yellow River has decreased with time[30], decoupling of TDS from the sediment trend has indicated the anthropogenic source and concentration effect[21] of increased $C_{TDS}$. Pattern V, exhibiting decreasing trends in both $Q$ and $C_{TDS}$, mostly affects rivers in semi-arid regions of North America, including the Colorado and Rio Grande, and rivers in Oceania (such as Murray). Pattern VI, with decreasing $C_{TDS}$ and increasing $Q$, only occurs for the Gila River in North America and the Tapajós River in South America, indicating the dilution effect in small basins[31]. Finally, Pattern IX with increasing trends in both $Q$ and $C_{TDS}$ is found to occur at a mere seven stations in North America, where severe anthropogenic perturbation is implied. Each of the nine [$Q$, $C$] co-varying patterns of runoff and $C_{TDS}$ is described by a representative large river (Supplementary Fig. 1); the underlying natural and human causes are summarized in Supplementary Table 1.

The [$Q$, $C$] co-varying patterns were also characterized for dissolved solids (DS) ions and river discharge in the global large rivers (Supplementary Fig. 2). Similar to the co-varying patterns of TDS, Patterns I, IV, and VII, each characterized by stable annual runoff, predominate (6.7–63%). Each solute ion exhibits distinct distributions of co-varying patterns across the six continents, reflecting differences in climatic, geological, hydrological, and human conditions. By examining the trends of individual DS ions, we identified several hotspots of anthropogenic stresses on global river systems (Supplementary Fig. 3). For example, increasing trends of $Ca^{2+}$, $Mg^{2+}$, and $Na^+$ concentration and annual runoff in the Mississippi River (Supplementary Fig. 3a–c) correlate with freshwater salinization, mainly caused by human activities such as agricultural irrigation, salt pollution (e.g., road deicers, sewage, potash), mining, and increased use of easily weathered minerals (e.g., lime in agriculture and concrete in construction)[3,11]. Increasing $K^+$ and $Cl^-$ concentrations at the Toudaoguai station of the Yellow River (Supplementary Fig. 3d, f) most likely resulted from saline return flows in the Qingtongxia

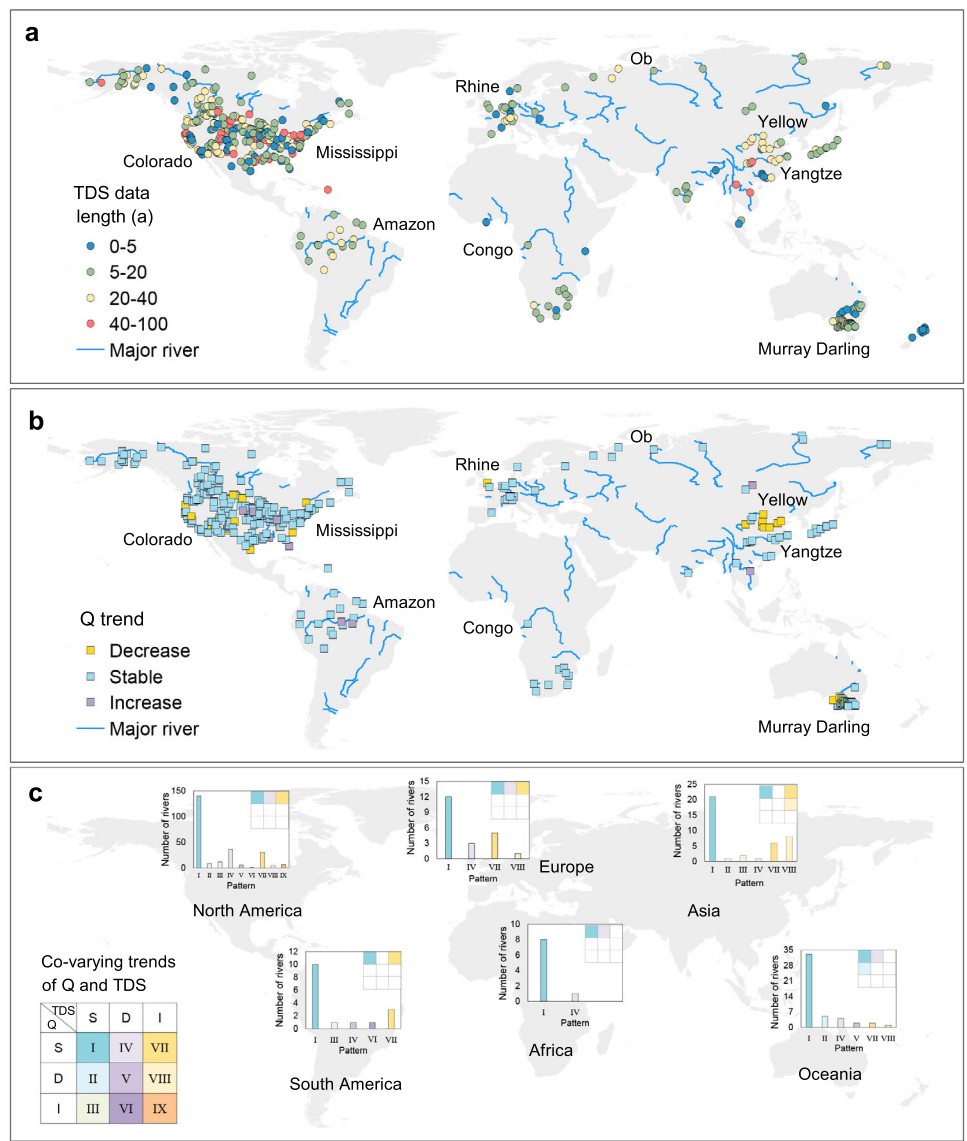

**Fig. 1 Global distribution of trends in total dissolved solids (TDS) and water discharge (Q). a** Global distribution of TDS data length in 149 world's large rivers. **b** Trends in Q for 600 hydrologic stations. **c** Number of rivers with different patterns of co-varying trends in Q and TDS in each of six continents. The insert Sudoku table and histograms display the three (or two) dominant patterns for each continent. All trends are significant at the 0.05 significance level.

and Hetao agricultural irrigation zones[32]. At Datong station of the Yangtze River, $SO_4^{2-}$ concentrations increased with water discharge (Supplementary Fig. 3e), owing to coal-burning and acid deposition in the basin[17,18]. In fact, the Chongqing–Guiyang region located in the upper Yangtze River basin is among the regions in the world most severely affected by acid deposition[18,19]. Surprisingly, the Ob River, one of the largest rivers in the Arctic zone, showed increasing $HCO_3^-$ concentration and river discharge (Supplementary Fig. 3g), driven by increased temperature, precipitation, permafrost thaw, agricultural liming, decreased acid deposition, and alteration of hydrology and vegetation[33,34]. With 22.9% of its basin covered by cropland and a population density of 9.51 people/km², anthropogenic stresses are stronger in the Ob River basin than in any other Arctic river basin[35]. Finally, decreasing DSi concentrations at Lee's Ferry station downstream of the Glen Canyon Dam in the Colorado River (Supplementary Fig. 3h) are attributed to decreasing silica-bearing sediment flux caused by the presence of the dam[20].

**Global fluxes of solutes to the oceans**. Combining our dataset with others[36,37] based on the COastal Segmentation and its related CATchment (COSCAT) concept[38], we estimated the global fluxes of solutes to the oceans (Fig. 2). Global TDS flux to oceans accounted for 6393 Mt/yr, about 68% greater than previous estimates of 3600–3800 Mt/yr[2,4,5]. Similarly, the estimated flux values increased significantly by 142% for $SO_4^{2-}$, 86% for $Na^+$, 81% for $Cl^-$, 71% for $Mg^{2+}$, 62.5% for $K^+$, 57% for $Ca^+$, and 22% for $HCO_3^-$, compared to published data (Supplementary Table 2). The proximity of the DSi flux to previous estimates (within 10%) partially validated the present calculations because natural environmental variables determined the most stable DSi fluxes[2]. Moreover, the predominant patterns with stable Q co-varying trends indicated that the accuracy of the estimated DS fluxes would be increased by using modeled discharge from Fekete et al.[36]. The increase in estimated global fluxes of other ions (especially $SO_4^{2-}$, $Na^+$, and $Cl^-$) is likely to have been related to the use of data of improved quality, which included the effects of recent and long-term anthropogenic disturbances.

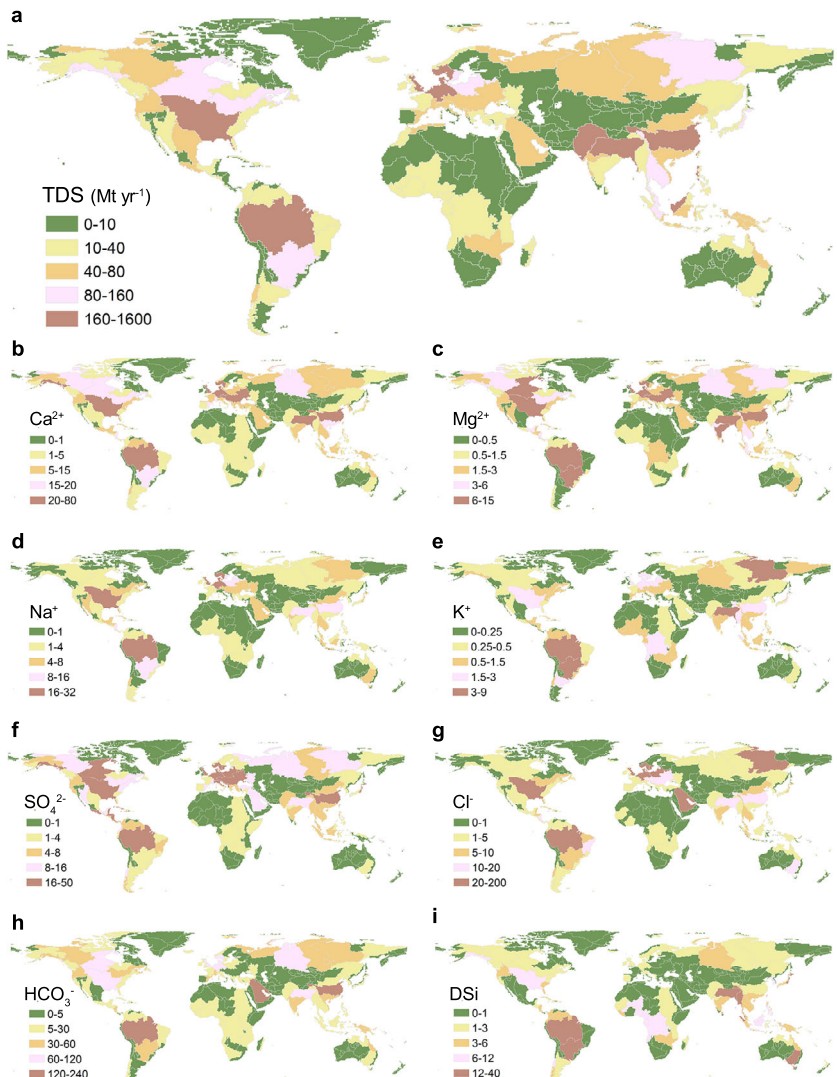

**Fig. 2 Annual flux of total dissolved solids (TDS) and major solutes from rivers to oceans. a** refers to TDS flux, **b–i** correspond to $Ca^{2+}$, $Mg^{2+}$, $Na^+$, $K^+$, $SO_4^{2-}$, $Cl^-$, $HCO_3^-$, and dissolved silica (DSi) flux.

We further identified the hotspots for global fluxes of DS to the oceans. TDS flux in the Mississippi River Basin (COSCAT-834) was extremely high (1560 Mt/yr), and other hotspots occurred in the Yangtze River (COSCAT-1326, 255 Mt/yr), the Mekong River (COSCAT-1329, 102 Mt/yr), and the Weser River (COSCAT-403, 203.5 Mt/yr), all due to coupled high TDS concentration and annual runoff. However, the Yellow River (COSCAT-1325, 68 Mt/yr), despite high $C_{TDS}$, had moderate TDS flux because of the relatively low river discharge (Fig. 2 and Supplementary Fig. 1h). The Amazon River (COSCAT-1104) exported only 473 Mt TDS annually to the Atlantic Ocean, even though annual runoff was extremely high. Moreover, the low TDS fluxes in the rivers of Africa (Supplementary Fig. 1a) and Oceania (Supplementary Fig. 1b) were due to stable TDS and low river discharge. Turning to specific ions, European rivers showed high fluxes of $Na^+$ and $Cl^-$, as also found by Milliman and Farnsworth[2], owing to intensive human activities (e.g., mining and agriculture). Magnesium fluxes were extremely high in the Indian Peninsula, South Asia (COSCAT-1337), resulting from the high magnesium content of water and soil in irrigation schemes, and leading to water quality deterioration and land degradation. In addition, high levels of bicarbonate fluxes were

exported by Arctic rivers (COSCAT-1307, 1308, and 1309), as also observed by Drake et al.[33].

**Formulation and global distribution of solute-induced river syndromes**. A solute-induced river syndrome represents a drastic change to a river system driven by extreme changes in dissolved solids concentration, pH, hardness, or alkalinity. This drastic change usually manifests itself as high/low contents of particular solutes, resulting from negative interactions between humans and the environment. Each solute-induced river syndrome, identified according to a certain threshold (see "Methods"), is characterized by specific symptoms, impacts, spatial distributions, and representative rivers (Table 1 and Fig. 3). In Supplementary Fig. 4, we further demonstrated the thresholds used to identify the solute-induced river syndromes for given temporal trends in solutes concentration.

Figure 4 presents the latitudinal distribution of mean annual DS concentrations under varying trends in $Q$. Stable, decreasing, and increasing trends in mean annual solute concentrations are identified in Supplementary Fig. 5. A previous study[10] of the global latitudinal distribution of natural riverine silica reported maximum concentrations in subtropical regions (around 20°N)

**Table 1 Characteristics of typical solute-induced river syndromes in the world's large rivers.**

| Syndrome (number of sites) | Symptom | Identification criteria | Impact | Continent | Representative river (site) | References |
|---|---|---|---|---|---|---|
| Salinization (2 sites) | Salt contents↑, TDS$^a$↑ | $\Sigma^+$ (Ca$^{2+}$ + Mg$^{2+}$ + Na$^+$+K$^+$) > 24 meq/L, increasing trend | Infrastructure corrosion; contaminant transport; safe drinking water; biodiversity | / | Rhine River (Maassluis) | 58 |
| Mineralization (33 sites) | | $\Sigma^+$ (Ca$^{2+}$+Mg$^{2+}$+Na$^+$+K$^+$) > 3 meq/L, increasing trend | | Central, Southeast of North America; West of Europe | South Saskatchewan River (Medicine Hat) | 26,59 |
| Desalinization (12 sites) | Salt contents↓, TDS↓ | $\Sigma^+$ (Ca$^{2+}$+Mg$^{2+}$+Na$^+$+K$^+$) <1.5 meq/L, decreasing trend | Drinking water supply; river-ocean ecosystem | Oceania | Murray River (Torrumbarry) | 60,61 |
| Acidification (8 sites) | HCO$_3^-$↓, pH↓, hardness↑ | pH <7; hardness/alkalinity > 1, increasing trend | Impacts on ecosystem and biodiversity; infrastructure corrosion | South America; Southeast of North America | Beni River (Rurrenabaque) | 62-66 |
| Alkalization (12 sites) | HCO$_3^-$↑, pH↑, hardness↓ | pH >7; hardness/alkalinity < 1, decreasing trend | Infrastructure damage, Impacts on irrigation areas, ecosystem, and biodiversity | Central North America; Northern Asia; Southern Africa | Groot Vis River (Matomelas Reserve) | 67,68 |
| Hardening (36 sites) | (Ca$^{2+}$ + Mg$^{2+}$)↑, cation↑ | Hardness (CaCO$_3$ mg/L) >120, increasing trend | Threat to soils and safe drinking waters; impacts on ecosystem | North America; Western Europe; South and East of Asia | Yellow River (Toudaoguai) | 32 |
| Softening (18 sites) | (Ca$^{2+}$ + Mg$^{2+}$)↓, cation↓ | Hardness (CaCO$_3$ mg/L) < 60, decreasing trend | Impacts on ecosystem and biodiversity | Central South America; Southeastern Oceania | Roanoke River (Randolph) | 69,70 |

$^a$Total dissolved solids.

but much lower concentrations in the boreal regions. Remarkably, Fig. 4 reveals that stations with maximum solute concentrations converge spatially to belts about 30–50ºN and 30–40ºS.

Although chemical weathering is accelerated in geologically active and tropical regions, minimum solute concentrations occur close to the equator (5ºS–5ºN) owing to the already deeply weathered environment in Brazilian and African shields located in the central tropics[39]. Low concentrations could be further attributed to tectonic stability (>>100 My) and high precipitation in these wet tropical regions[40,41]. It appears that solute-induced river syndromes occur as combined consequences of changes in both solutes themselves and $Q$ (Fig. 4 and Supplementary Fig. 5), accompanied with data-driven trends at syndrome sites clustered in three "hot-belts" (30°–50ºN, 30°–40ºS, and 5ºS–5ºN).

**Natural and anthropogenic impacts on changes in solutes with river syndromes.** In 1970, Gibbs[13] proposed a diagram for interpreting the natural genesis of river water chemistry (atmospheric precipitation, rock weathering, and the evaporation–crystallization process). Figure 5a, b shows the Gibbs diagram derived from the dataset containing 60 gauge stations with solute-induced river syndromes, determined as average cation and anion ratios of dissolved ions (Ca$^{2+}$, Na$^+$, Cl$^-$, and HCO$_3^-$) plotted against TDS. Almost all the solute-induced river syndrome sites are scattered in the rock-weathering zone, except for several in the evaporation–crystallization and precipitation zones. As shown in Supplementary Fig. 6, sites with abundant precipitation that diluted solutes leading to decreasing $C_{TDS}$ trends lie in the precipitation zone. The majority of syndrome sites with increasing $C_{TDS}$ trends occur in the rock-weathering zone, indicating the possible influence of intense natural and anthropogenic impacts. Despite the notable variation trend of $C_{TDS}$, the majority of sites affected by solute-induced river syndromes exhibit a stable trend in $Q$ (Fig. 5a, b), highlighting the importance of changes in solutes themselves. Conventionally, the Gibbs diagram has been used to explain natural drivers of river water composition solely for areas with low human interference. By incorporating the $Q$–$C$ trends of solute-induced river syndrome sites in the world's large rivers, the present Gibbs diagram provides additional information potentially relevant to recent anthropogenic disturbance. For instance, irrigation intensifies rock chemical weathering[7,11]; dam operations increase evaporation and humidity, altering the rainfall distribution while enhancing precipitation over the whole basin[42]; and the urban heat island effect due to population growth and accelerated urbanization influences the global climate[43]. Increasingly, intense anthropogenic perturbation is therefore altering the natural processes that affect dissolved solids in rivers.

Lithology[12], climate[15], land use and land cover[26], agricultural activities[7], population[6], and dam construction[25] are key environmental factors controlling solute concentrations. We, therefore, selected carbonate sedimentary and acid volcanic (or igneous) rocks, arid and temperate climate classification, global irrigation area, and urban land-use type, as primary influence factors for solutes, noting their significant levels from redundancy analysis (RDA) listed in Supplementary Tables 3–5. Extraction of these factors was based on the Global Lithological Map v1.0[44], Köppen–Geiger climate classification[45], Global Land Cover by National Mapping Organizations (GLCNMO)[46], and Global Map of Irrigation Areas version 5[47], as explained in Supplementary Table 6 and Supplementary Methods.

Figure 5c, d depicts the latitudinal distribution (per 10° latitudinal belt) of temperate climate classification and urban land-cover classification. Supplementary Fig. 7 (per 5° latitudinal belt) presents latitudinal distributions of carbonate sedimentary rock, acid volcanic rocks, arid climate classification, and global irrigation. The largest proportions of carbonate sedimentary and acid igneous rocks are

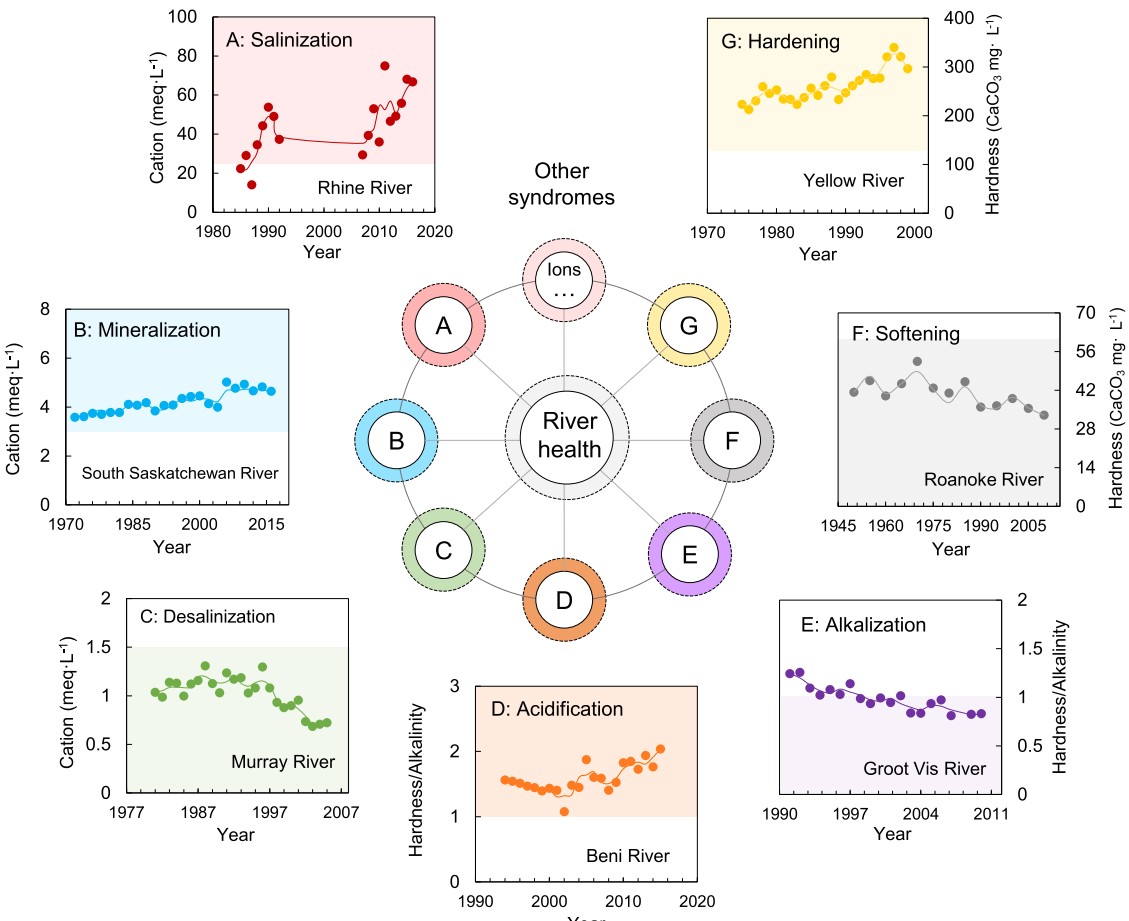

**Fig. 3 River syndromes induced by extreme changes in solutes.** A–G represent seven river syndromes associated with ion concentration. The colored area indicates that the river is experiencing a specific syndrome as the solute metric exceeds a certain threshold. The central open circle refers to the ion concentration range when the river ecosystem is in a healthy state. Colored annuli refer to ion concentrations that exceed prescribed thresholds, causing river syndromes. Colored dots in (A–G) denote measured data, and colored lines indicate the three-year moving average.

distributed around 30–50°N (Supplementary Fig. 7a, b). Spatial variations in $HCO_3^-$ and $Ca^{2+}$ concentrations relate to the carbonate sedimentary rock distribution[48], whereas the DSi concentration correlates strongly with the percentage of acid volcanic rocks[48]. According to the Köppen–Geiger climate classification, the temperate climate that promotes optimum weathering is mainly distributed around 30–40°N and 30°S (Fig. 5c). Arid climate, also prominent around 30–40°N and 30°S (Supplementary Fig. 7c), further strengthens anthropogenic impacts (e.g., in North America[49]). The latitudinal distributions of irrigation (Supplementary Fig. 7d) and urbanized land cover (Fig. 5d) indicate severe anthropogenic interference[2,6] is occurring around 30–40°N, providing a partial interpretation of the maxima in $Na^+$ and $Cl^-$ solute ions in this region. Note that over half of both anthropogenic terrestrial and aquatic emissions occur between 20°N and 66°N[50]; anthropogenic impact on dissolved solids should be significant in this region.

RDA further differentiates natural and anthropogenic factors controlling solutes relevant to river syndromes (68 sites, Supplementary Fig. 8, and Supplementary Tables 3–5), particularly at 58 sites where syndromes occurred in the critical latitudinal belts (Fig. 5e, f). All natural factors displayed notable correlations with DSi ($P < 0.05$), except ice. Among the anthropogenic factors, irrigation presented the strongest correlation with the solutes considered ($P < 0.05$).

Arid climate correlated positively with $Na^+$, $SO_4^{2-}$, and $Cl^-$, but negatively with $HCO_3^-$ and DSi (Fig. 5e and Supplementary Fig. 8a). Oppositely directed correlations are obtained between tropical

climate types and the different solutes. Dilution of $Na^+$, $SO_4^{2-}$, and $Cl^-$ ions would have occurred in areas of abundant precipitation, corresponding with Meybeck's findings[51]. In addition, rising river discharge could enhance dilution, leading to lower DS concentrations close to the equator (Fig. 4). In hot and humid regions, the $HCO_3^-$ and DSi concentrations increase due to the rock-weathering-dominant sources. Temperate and cold (polar) climate types exhibit significant correlations with DSi, $Na^+$, $Cl^-$, or $Ca^{2+}$, $SO_4^{2-}$, indicating the different weathering and dilution (or concentration) mechanisms for the solutes. For instance, the positive correlation of DSi with tropical/temperate climate and its negative correlation with cold (polar)/arid climates illustrates the greater proportion of siliciclastic sedimentary and igneous rocks in mid-latitude zones[44], in accordance with Supplementary Fig. 7.

In view of the close relationships of sedimentary and igneous rocks with solutes (Fig. 5e and Supplementary Fig. 8a), RDA was conducted for sub-classified rock types at sites with solute-induced river syndromes (Supplementary Fig. 9 and Supplementary Table 5). Carbonate sedimentary type correlated positively with $Ca^{2+}$, $SO_4^{2-}$, and $HCO_3^-$, and negatively with $Na^+$, $Cl^-$, and DSi. Siliciclastic sedimentary type correlated positively with $SO_4^{2-}$, $Na^+$ and $Cl^-$. Acid and basic volcanic types exhibited positive correlations with DSi and $HCO_3^-$, indicating that DSi was primarily derived from the weathering of igneous rocks. However, these two types of rocks presented negative correlations with $SO_4^{2-}$, suggesting that $SO_4^{2-}$ was mainly derived from weathering of siliciclastic and carbonate sedimentary rocks containing sulfates.

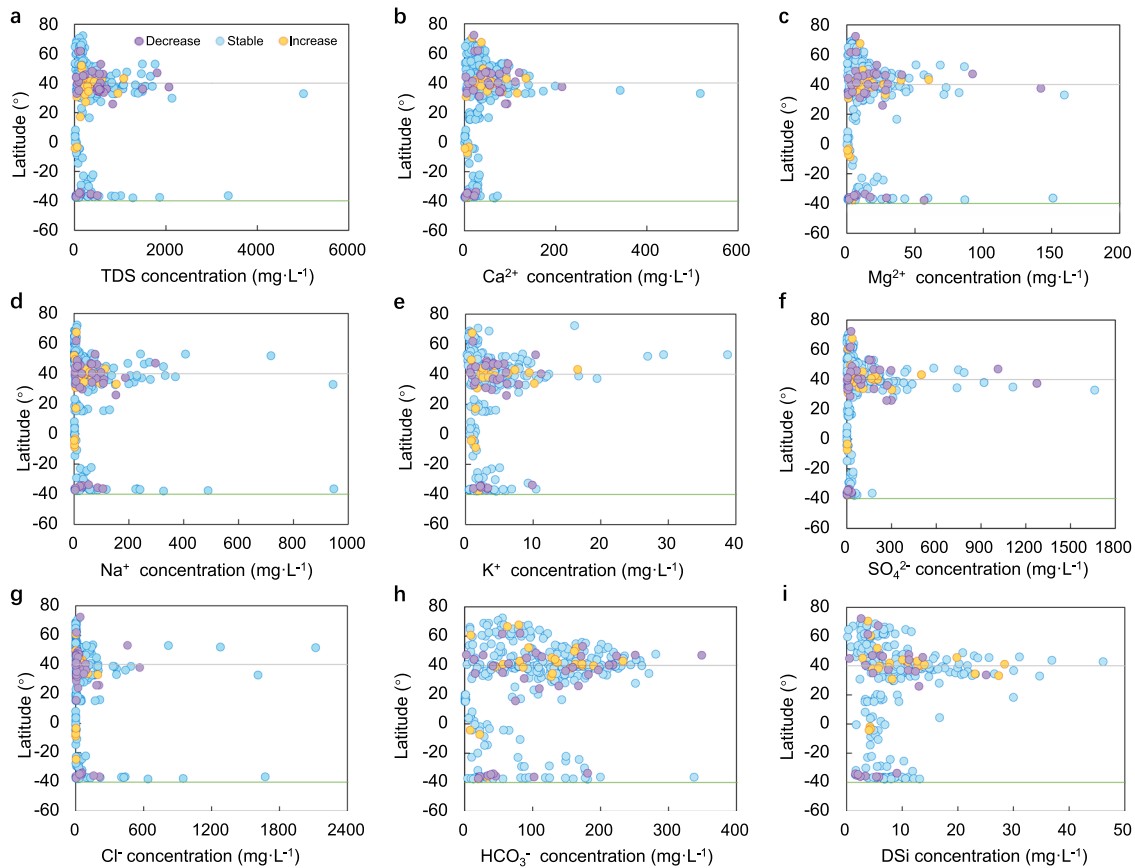

**Fig. 4 Latitudinal distribution of mean annual solute concentrations under varying trends in water discharge (Q). a** Total dissolved solids (TDS). **b** $Ca^{2+}$. **c** $Mg^{2+}$. **d** $Na^+$. **e** $K^+$. **f** $SO_4^{2-}$. **g** $Cl^-$. **h** $HCO_3^-$. **i** Dissolved silica (DSi). The 476 dots in purple, blue, and yellow respectively represent rivers with decreasing, stable, and increasing trends in annual runoff.

For sites with solute-induced river syndromes, RDA results (Supplementary Table 4) revealed the anthropogenic stresses due to agricultural activities. RDA plots obtained for anthropogenic factors (Fig. 5f and Supplementary Fig. 8b) indicated positive correlations between $Na^+$, $Cl^-$ and irrigation, agriculture, population, in accordance with findings from a previous study of the Mekong River[6]. Urbanization and degree of regulation (DOR) correlated positively with $SO_4^{2-}$ and $Na^+$, and negatively with $HCO_3^-$ and DSi, implying that $SO_4^{2-}$ and $Na^+$ may be effective indicators of the influence of urbanization and dam regulation. Here, increased $SO_4^{2-}$, $Cl^-$, and $Na^+$ levels may be associated with severe anthropogenic disturbance (e.g., urbanization and irrigation), in agreement with other studies[2,3,32]; whereas natural processes (e.g., weathering) played a vital role in controlling the concentrations of $Ca^{2+}$, DSi, and $HCO_3^-$.

Sites experiencing solute-induced river syndromes in the three critical latitudinal belts exhibited the highest constrained variances (Fig. 5e, f), reflecting the intense impact of environmental factors on water quality. A holistic understanding of global solute-induced river syndromes is of fundamental importance for identifying and prioritizing regional management requirements and local amelioration strategies. The top priority is to offer advice regarding the time when a solute-induced river syndrome is likely to occur or has occurred. To facilitate practical management, a warning line could be set for each of the solute-induced river syndromes. Using a rule of thumb approach[7], alarm values are identified in terms of the solute metrics as they approach 90% of the prescribed thresholds for solute-induced river syndromes (Supplementary Fig. 4). The trends in solute concentration suggest the developing directions towards or away from the syndromes, while the co-varying trends in both runoff and

solute concentration provide potentially important information on major causes of river syndromes associated with solutes evolution.

Given the significant impacts of various solute-induced river syndromes on water quality, contaminant transport, biodiversity, and ecosystem function (Table 1), precautionary countermeasures should be implemented for effective control of syndrome development in alarmed rivers. For instance, effective regulation of river flow, waste emissions, and agricultural runoff is necessary to control water hardening in North America, Western Europe, and South and East Asia. Improved management practices are required to prevent severe salinization in central and southeast North America and west Europe where population density is high. Knowledge of the inherent susceptibility of river systems to solute syndrome development should be utilized in regional decision-making processes, e.g., the presence of carbonated rocks encourages acidification, and semi-arid regions are very sensitive to salinization. Moreover, sustainable strategies should be made to alleviate climate influence on chemical weathering of rocks (particularly by $HCO_3^-$ and DSi), with particular attention to mitigation of river syndromes in the process of urbanization and agricultural irrigation due to extreme changes in solute concentrations (especially $Na^+$, $SO_4^{2-}$, and $Cl^-$).

The strict data requirements of simultaneous runoff (Q) and solute concentrations (C) limited the data available for this study. Considering the potential influence of the location of monitoring sites on the spatial distribution of solute-induced river syndromes, future effort should be made to enrich the dataset with greater coverage through enhanced long-term monitoring, particularly for rivers in areas with scarce data. To compensate for the lack of available data, biogeochemical predictive models

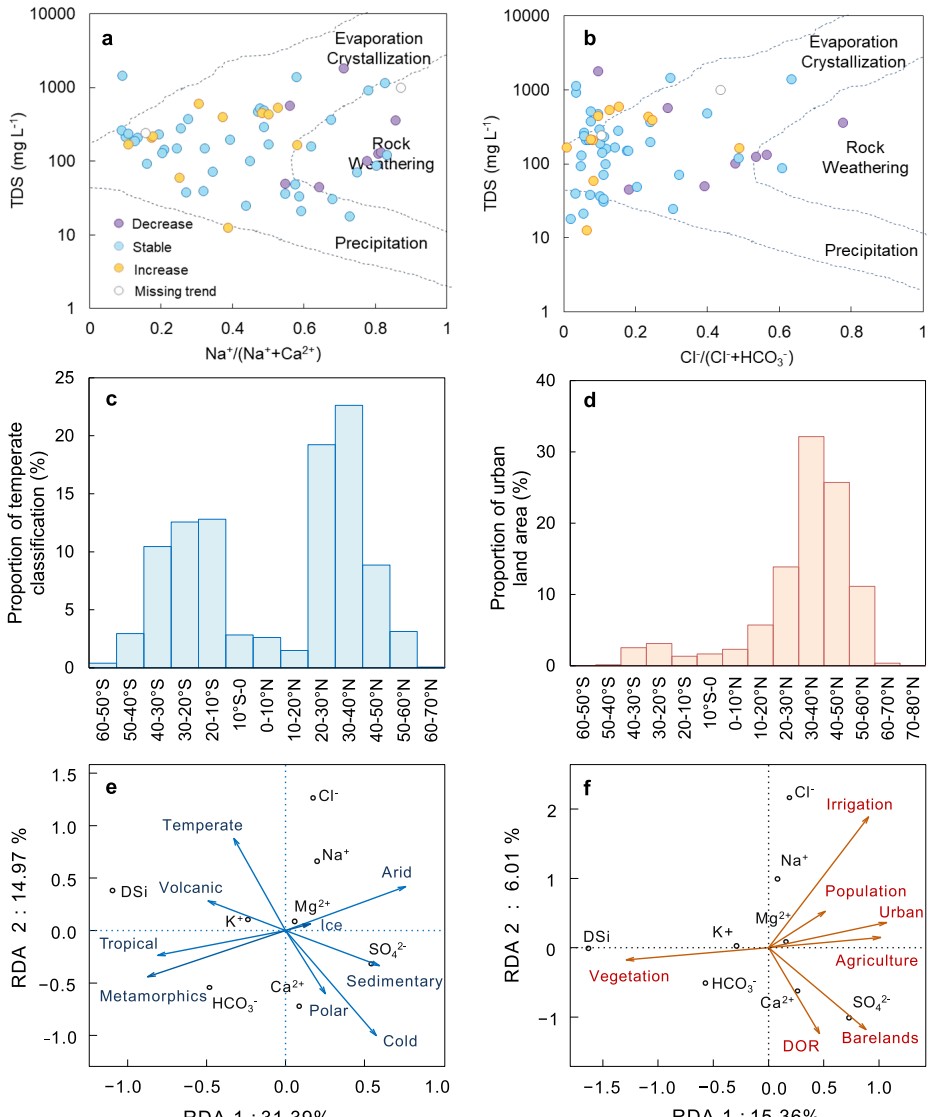

**Fig. 5 Interpretation of natural and anthropogenic impacts on changes in solutes with river syndromes. a, b** Gibbs model for stations with solute-induced river syndromes under varying trends in water discharge ($Q$). The 60 dots in purple, blue, and yellow respectively represent decreasing, stable, and increasing trends in the annual runoff. **c, d** are latitudinal distributions of proportional areas of temperate climate type and urban land area. **e, f** are redundancy analysis (RDA) results for the influence on dissolved solids of natural factors (reclassification for Köppen climate and lithology) and anthropogenic factors (irrigation, population, degree of regulation (DOR), and detailed reclassification of land cover), in critical latitudinal belts.

should be developed in conjunction with global monitoring campaigns, to link the solute-induced river syndromes with the natural conditions and human disturbances of the basins, and predict their future evolution under different scenarios.

In summary, using a newly established global database containing annual runoff and solute concentration information at 600 stations in 149 large rivers, we examined the geographical spread of seven solute-induced river syndromes (salinization, mineralization, desalinization, acidification, alkalization, hardening, and softening) in the world's large rivers. In general, a stable [$Q$, $C$] trend predominated the nine representative co-varying trends across the global rivers. However, significant increases occurred in total dissolved solids (68%), chloride (81%), sodium (86%), and sulfate (142%) fluxes from rivers to oceans worldwide, leading to typical solute-induced river syndromes on exceeding specific thresholds. Our study revealed that the syndromes were mostly concentrated in two latitude zones (about 30–50°N and 30–40°S, respectively) which experience severe rock weathering accelerated

by anthropogenic disturbances such as rapid urbanization and agricultural irrigation. Despite acidification being observed close to the equator (5°N–5°S), such syndromes were mainly exacerbated by extreme dilution of solutes vital for river health, such as $HCO_3^-$ in the rivers of South America. This study provides insight into global changes in major river solutes and highlights their importance to the geochemical cycle, and is of significance to the water security management of large river ecosystems.

## Methods

**Data source.** We compiled time series (1915–2018) of annual runoff ($Q$) and concentrations ($C$) of solutes for analysis of co-varying trends in ($Q$–$C$) and solute fluxes at 600 monitoring stations located in 149 rivers (basin area ≥1000 km$^2$) across six continents. Briefly, part of the data was collected and compiled from publicly available databases including Arctic Great Rivers Observatory, Canada's National Water Data Archive, Global Environment Monitoring System for Water (GEMS), Hydrological Yearbooks of the People's Republic of China, the Observation Service SO HYBAM, and United States Geological Survey. The remaining data were taken from the open literature and websites (Supplementary Tables 7 and 8 and Supplementary Methods provide details of all the data sources). The stations

are mostly located in North America (54.8%), followed by Asia (14.8%), Oceania (12.2%), Europe (8.2%), South America (7.5%), and Africa (2.5%) (Fig. 1). The 149 rivers collectively drain 46 million km$^2$ of watersheds. For dissolved solids, 64–73% of the stations have a record length no less than 10 years (Supplementary Table 9) with the longest record up to 93 years. Data quality was ensured by detecting and removing 1888 outliers (2.3% of all data) in the time series of solutes. In a subset of our data (6124 samples with available concentrations for all seven charged ions of interest during the same period at the same monitoring station), 89% of the data points met the normalized inorganic charge balance criteria (−10%~10%) for cations and anions, guaranteeing the data were of high quality. The Supplementary Methods provides a detailed description of the database compilation and quality assurance procedures.

**Estimation of solutes fluxes.** A hybrid approach was used to calculate the annual fluxes of solutes. When the measured annual average concentration of a specific solute was available for a given year at a specific station, the annual flux was determined as the product of the annual average concentration with annual runoff. When measured data was unavailable, the LOAD ESTimator (LOADEST) model[52] was used to estimate annual average concentrations and loads for the available daily or monthly river flow. For 15,702 available data points, the observed and calculated annual fluxes were in good agreement ($R^2 > 0.9$, Supplementary Fig. 10), suggesting the reliability of the hybrid approach that exploited both observed and predicted datasets. The COastal Segmentation and CATchment (COSCAT)[38] concept was used to estimate the fluxes of solutes to the sea based on average data calculated from our datasets, ion concentration datasets by Meybeck and Ragu[37], and the estimated discharge of each COSCAT[36] (Supplementary Methods).

**Trend analysis.** We used non-parametric trend analysis TFPW–MK (Trend-Free Pre-Whitening Mann–Kendall)[53,54] derived from the Mann–Kendall (M–K) method to determine any significant trends in the time series of $C$, $Q$, and flux ($F$) with ≥5-year record. The "modifiedmk" package of R software performed the TFPW–MK test. Detailed procedures for trend analysis are given in Supplementary Methods.

**Determination method for global solute-induced river syndromes.** Three solute-induced river syndromes were identified following Meybeck's classification[51] for different solute trends: salinization with $\Sigma^+$ ($\Sigma^+ = C_{Ca}^{2+} + C_{Mg}^{2+} + C_{Na}^+ + C_K^+$) > 24 meq/L indicating an increasing trend, mineralization with $\Sigma^+ > 3$ meq/L indicating an increasing trend, and desalinization with $\Sigma^+ < 1.5$ meq/L indicating a decreasing trend. To determine whether a river experiences acidification or alkalization, both pH and alkalinity (or acidity) must be considered[18,21,33]. Here, we calculated the ratio of hardness to alkalinity, and extracted mean pH values for rivers from the GEMS database[55]. A hardness to alkalinity ratio >1 suggested an acid input from anthropogenic sources[56]. The acidification syndrome was deemed to occur in a river when pH <7 and the ratio of hardness to alkalinity >1 indicating an increasing trend. The converse indicated an alkalinity syndrome. The hardening syndrome in a river corresponds to an increasing trend in hardness >120 mg/L, whereas the softening syndrome corresponds to a decreasing trend in hardness <60 mg/L. The Supplementary Methods give further details of the methodology used to identify the different solute-induced river syndromes.

**Extraction of typical natural and anthropogenic environmental factors.** Lithology[44] (sedimentary, volcanic and metamorphic rocks, and ice) and Köppen–Geiger climate classification[45] (reclassification of arid, temperate, tropical, cold, and polar climates) were selected as natural factors. Primary anthropogenic environmental factors were land cover (reclassification of vegetation, urban, agriculture, and bare land), irrigation, dam, and population. The Supplementary Methods describe the approach used to extract information on the environmental variables from the metadata; Supplementary Table 6 lists the detailed environmental information.

**Reporting summary.** Further information on research design is available in the Nature Research Reporting Summary linked to this article.

## Data availability

Data sources for concentrations of dissolved ions and runoff in the world's large rivers are available within the paper and its supplementary information file. Data on solutes concentrations ($C$), annual runoff ($Q$), the co-varying trend in $Q$ and $C$, and the identified solute-induced river syndrome if applicable, at each river station have been deposited on figshare [https://doi.org/10.6084/m9.figshare.14910399][57]. Source data are provided with this paper.

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

## Acknowledgements

Financial support is from the National Natural Science Foundation of China (Nos. 51721006 and 91647211).

## Author contributions

J.R.N. designed the research. J.W. established the dataset and performed the research with help of the co-authors. J.W., J.R.N., and N.X. wrote the paper. W.Z., Y.C.W., and A.B. contributed new ideas and information. All of the authors contributed to the interpretation of the findings.

## Competing interests

The authors declare no competing interests.
