## [Peer Review File · Nature Communications]

Global syndromes induced by changes in solutes of the world's large riversREVIEWER COMMENTS

Reviewer #2 (Remarks to the Author):

Global Syndromes induced by changes in solutes of the world's largest rivers, Wu et al.
Nature Communications

I had the opportunity to review the manuscript Global Syndromes induced by changes in solutes of the world's largest rivers by Wu et al. In this paper, the authors evaluate a global data set of cations and anions to assess the frequency of "river syndromes" including salinization, mineralization, desalinization, acidification, alkalization, hardening and softening in large rivers. The authors document large increases in the flux of multiple solutes. Changes in solutes are attributed to anthropogenic forcings including urbanization and irrigation. The comprehensive data set that the authors have put together is impressive and the evaluation of these trends over large scales is a meaningful contribution.

It is important to compare/contrast this work to that of Kaushal and Meybeck (which, to the author's credit, are appropriately acknowledged and cited). Although there have been many papers looking at global trends in these solutes, I focus on Kaushal and Meybeck as these really develop and quantify the concept of riverine syndromes. In Kaushal et al (2018; PNAS) many of these syndromes are named through the evaluation of many similar solutes. So, the advancement here is mapping these trends at the global scale compared to Kaushal's continental (USA) scale. Yet the temperate and N. American bias cause for much overlap. At the same time many of these syndromes are named in Meybeck 2003 where river syndromes are evaluated in the context of the Anthropocene. Here the advancement is to simultaneously evaluate co-varying trends in discharge (although Meybeck does discuss the widespread reduction in riverine flows due irrigation).

The vast majority of gauged and evaluated rivers, however, (83%) indicate a stable trend in Q. Because of this, the manuscript and logic are difficult to follow as the authors spend considerable text walking the reader through each of the nine syndromes and numerous site-specific examples as opposed to broad generalization of the data. Yet, Q doesn't seem to drive much of the variation. For example, in Figures 3 and 4 sites do not appear to cluster by variation in trends of Q. So, what does this add to the story? The answer to this important question is not overly clear to this reviewer.

One major conclusion of the paper that does not seem robust is that the syndromes appear to be concentrated around latitudes 40N and 40S. This conclusion is a clear artefact of the data. First, the majority of the data set is from the temperate region, so the conclusion is not because of something unique in these regions but rather that the data are clustered around this zone. I am not discounting the fact that the human footprint is significant along 40N, but it is not possible to state confidently that this is where the syndromes are most prevalent when it is the distribution of the data driving the pattern (see histograms). Second, the similar conclusion regarding 40S is difficult to accept. 40S primarily crosses the ocean and over little continental landmass. The prevalence of syndromes at 40S are due to the heavily sampled Murray-Darling system in Australia. So the 40S pattern is due to one site. Because of these two sampling artefacts, the global scale conclusions need to be tempered.

Reviewer #3 (Remarks to the Author):

This is a really compelling study of longterm changes to solute loads (concentrations and concentrations*discharge) for over 100 major river basins around the world. Such data are challenging to collect in a single place and analyze in a reasonable manner. Clearly the authors have worked diligently to do this here and have a very strong analysis of such data presented here. The novelty of the data, the analyses, and the findings warrant publication, in my opinion. However, it would be helpful if the following concerns were addressed so that the most accurate and appropriate

message is communicated to the scientific community (and others).

Moderate Concern:

1) The use of the term 'syndrome' for these sort of long-term changes is novel. I have not seen it used with respect to changes in river solutes before. I believe that the authors are using it appropriately here given their citation of other uses of 'syndrome' in environmental contexts. However, because this is the first use of it for river solutes, I would advocate for being very careful to articulate exactly what this term means in the context of river solutes. What sort of 'certain thresholds' are needed for these to have been identified for rivers? It seems to me there's a greater analysis of temporal trends here (for each basin) rather than an analysis of specific threshold identification and then exceedance. It would be helpful to resolve this for accuracy of message.

2) Related... One thing I have some trouble figuring out is perhaps related to the use of the term 'syndrome' and the varying changes in solute loads around the world. That is, I typically associate the term 'syndrome' to indicate a single sickness/challenge, etc. that affects many people, but in all cases it can be diagnosed the same way. In this case, the river syndromes are quite varied. In some cases salinization is occurring, in others, freshening (the opposite) is occurring. Further, the authors note that the causes of these different trajectories are somewhat unique across river basins too (i.e., some are from changes in weathering rates, some are from anthropogenic activities). I am not sure how to rectify this except perhaps to (in combination with making edits to reply to #1 above) be explicit that the syndrome is purely about long term changes in solute concentrations and/or loads. Maybe re-name to always be referred to as 'solute syndrome' which allows for any solutes to be included?

3) The authors recommend attempts to control or curb some activities that may be contributing to syndromes (lines 292-300), but I wonder if they can offer advice about *when* one can determine that a syndrome has occurred or is occurring. Any trend over a few years can be spurious, and yet at the other end of the temporal spectrum, decades of data collection would mean that the syndrome has occurred for too long. How can we detect these events early (and robustly)?

Minor Concerns

1) Perhaps I'm missing this but the emphasis in a few places of the big changes in river basins from 40S to 40N latitude has me thinking that this is also the region of the globe with the greatest human population. I'm not sure that that's true, but it sure seems like it would be. Is there an easy way to figure this out and then perhaps make that correlation (I'm not necessarily advocating for a statistical analysis) here? I see the relationship to urban and agricultural lands in this range of the globe, but ultimately isn't this also about people being particularly densely confined to this region compared to the rest of the globe?

2) line 133 - not all permafrost is 'ice' or 'ice-rich' so it's better to say that permafrost 'thaws' rather than melts.

Response to Reviewers' Comments

Manuscript title: Global syndromes induced by changes in solutes of the world's large rivers

Reference Number: NCOMMS-21-04236-T

Response to Reviewer #1

Comment No.1: I had the opportunity to review the manuscript Global Syndromes induced by changes in solutes of the world's largest rivers by Wu et al. In this paper, the authors evaluate a global data set of cations and anions to assess the frequency of "river syndromes" including salinization, mineralization, desalinization, acidification, alkalization, hardening and softening in large rivers. The authors document large increases in the flux of multiple solutes. Changes in solutes are attributed to anthropogenic forcings including urbanization and irrigation. The comprehensive data set that the authors have put together is impressive and the evaluation of these trends over large scales is a meaningful contribution.

Response: We thank the reviewer for their insightful suggestions and comments, which we believe have led to a greatly improved manuscript.

Comment No.2: It is important to compare/contrast this work to that of Kaushal and

Meybeck (which, to the author's credit, are appropriated acknowledged and cited). Although there have been many papers looking at global trends in these solutes, I focus on Kaushal and Meybeck as these really develop and quantify the concept of riverine syndromes. In Kaushal et al (2018; PNAS) many of these syndromes are named through the evaluation of many similar solutes. So, the advancement here is mapping these trends at the global scale compared to Kaushal's continental (USA) scale. Yet the temperate and N. American bias cause for much overlap. At the same time many of these syndromes are named in Meybeck 2003 where river syndromes are evaluated in the context of the Anthropocene. Here the advancement is to simultaneously evaluate co-varying trends in discharge (although Meybeck does discuss the widespread reduction in riverine flows due irrigation).

Response: Thank you very much. We fully agree with the Reviewer's comments about the advancements of the present study, through highlighting the global distribution of solutes and co-varying trends in both solutes and discharge of the world's large rivers. As the referee noted, our original manuscript cited the pioneering works of Kaushal (*Kaushal, S. S., et al. Freshwater salinization syndrome on a continental scale, PNAS, 2018, 115, E574-E583*) and Meybeck (*Meybeck, M. Global analysis of river systems: from Earth system controls to Anthropocene syndromes, Philos. Trans. R. Soc. London, 2003, 358, 1935-1955*). In the revised manuscript, we have further stressed the significance of these works which are pivotal to the development and quantification of the concept of riverine syndromes. The revised sentences read as follows:

“In the early 2000s, Meybeck (2003) defined river syndromes as negative feedbacks of river systems to human activities, primarily including river fragmentation, flow regulation, water-sediment imbalance, and chemical and microbial contamination processes.” (Lines 62-65 in the revised main text, the same below)

“More recently, Kaushal et al. (2018) developed the concept of freshwater salinization syndrome which links salinization and alkalization processes along hydrologic flow paths from small watersheds to coastal waters. Using long-term time series of specific conductance, pH, alkalinity, and base cation concentrations, Kaushal et al. investigated freshwater salinization syndromes in hundreds of stream and river sites throughout the continental United States.” (Lines 69-74)

“Based on a unique dataset of major solutes (Ca^{2+} , Mg^{2+} , Na^+ , K^+ , SO_4^{2-} , Cl^- , HCO_3^-) and dissolved silica derived from 600 gauge stations in 149 rivers (basin areas $\geq 1,000$ km^2), we proposed a framework of notable solute-induced river syndromes (salinization, mineralization, desalinization, acidification, alkalization, hardening, and softening) which could be identified using thresholds of solute concentrations and associated trends in the world’s large rivers. Our study provides a dynamic perspective by which to determine the occurrence of solute-induced river syndromes and how they develop with co-varying trends in runoff and solutes. This is of great significance in the maintenance of river ecosystem health.” (Lines 80-89)

Comment No.3: The vast majority of gauged and evaluated rivers, however, (83%) indicate a stable trend in Q . Because of this, the manuscript and logic are difficult to

follow as the authors spend considerable text walking the reader through each of the nine syndromes and numerous site-specific examples as opposed to broad generalization of the data. Yet, Q doesn't seem to drive much of the variation. For example, in Figures 3 and 4 sites do not appear to cluster by variation in trends of Q . So, what does this add to the story? The answer to this important question is not overly clear to this reviewer.

Response: Thank you very much. The Reviewer raises an important concern regarding why we investigated the co-varying trend in annual runoff (Q) and solute concentration (C). In fact, the nine patterns for co-varying trends essentially reflect the relative contributions of water and the solutes themselves to solute fluxes in the evolution of water chemistry and biogeochemical cycle in a river system. The co-varying trends in Q - C not only provide a dynamic perspective to determining when the seven solute-induced river syndromes occur under certain thresholds and how the syndromes then develop, but are also helpful to interpret how these syndromes can arise from the changes in both solutes themselves and river runoff. As Q is the primary flux transported by rivers, its trend plays a crucial role in the evolution of solutes, e.g. through enrichment or dilution. For rivers with a stable trend in Q (about 83% of the gauged and evaluated rivers in the present study), the river syndromes are primarily controlled by changes in solutes rather than Q , and so the causal analysis should be focused on natural (e.g. most notably rock weathering) and human drivers of changes in the solutes themselves. For rivers with increasing or decreasing trends in Q (about 17% of the gauged and evaluated rivers in the present study), the river

syndromes are the combined consequences of changes in both solutes themselves and Q due to climate change and human activities. For clarity, rivers with stable, increasing, and decreasing trends in Q are distinguished by colored dots in Figs. 4 and 5 of the revised manuscript. (Corresponding latitudinal distributions of mean annual solute concentrations under varying trends in solutes themselves are given in revised Supplementary Figs. 5 and 6).

Revised Fig. 4 Latitudinal distribution of mean annual solute concentrations under varying trends in Q . a, TDS. b, Ca^{2+} . c, Mg^{2+} . d, Na^+ . e, K^+ . f, SO_4^{2-} . g, Cl^- . h, HCO_3^- . i, DSi. The 476 dots in purple, blue, and yellow respectively represent rivers with decreasing, stable, and increasing trends in annual runoff.

Revised Fig. 5a-b Gibbs model for stations with solute-induced river syndromes under varying trends in Q . The 60 dots in purple, blue, and yellow respectively represent decreasing, stable, and increasing trends in annual runoff.

In the revised manuscript, we rewrote the corresponding text in a more logical way in order to discuss the seven solute-induced river syndromes and their major drivers with respect to the nine co-varying trends in Q - C based on the data analysis, particularly the role of stable, increasing, and decreasing trends in Q . (Lines 93-94, 102-103, 104-105, 195-197, 206-210, 223-225, 227-229)

Comment No.4: One major conclusion of the paper that does not seem robust is that the syndromes appear to be concentrated around latitudes 40N and 40S. This conclusion is a clear artefact of the data. First, the majority of the data set is from the temperate region, so the conclusion is not because of something unique in these regions but rather that the data are clustered around this zone. I am not discounting the fact that the human footprint is significant along 40N, but it is not possible to state

confidently that this is where the syndromes are most prevalent when it is the distribution of the data driving the pattern (see histograms). Second, the similar conclusion regarding 40S is difficult to accept. 40S primarily crosses the ocean and over little continental landmass. The prevalence of syndromes at 40S are due to the heavily sampled Murray-Darling system in Australia. So the 40S pattern is due to one site. Because of these two sampling artefacts, the global scale conclusions need to be tempered.

Response: Thank you very much. We agree that the global-scale conclusions in the present manuscript need to be tempered, in particular that data availability would influence the conclusion that the syndromes appear to be concentrated about latitudes 40°N and 40°S. Of course, the strict data requirements of simultaneous runoff (Q) and solute concentrations (C) highly limited the data available to us, even though we spent six years collecting data that qualified for the present study. The majority of data we could access were indeed from temperate regions; for example, the 56,136 monitoring sites within 30-50°N and 30-50°S account for 71.35% of the total number of sites with monitored mean annual solute (Ca^{2+} , Mg^{2+} , Na^+ , K^+ , SO_4^{2-} , Cl^- , HCO_3^- and dissolved silica) concentrations. Our dataset (established to date) indicated that the syndromes appear to be concentrated in belts at latitudes 30-50°N and 30-40°S which experience severe rock weathering accelerated by anthropogenic disturbances such as rapid urbanization and agricultural irrigation (see also the Figures below). Considering the potential influence of the location of monitoring sites on the spatial distribution of the syndromes, future effort should be made to enrich the dataset with

greater coverage through enhanced monitoring, particularly for rivers in areas with scarce data. In the revised manuscript, we reworded relevant sentences in the Abstract and Results. Moreover, we provided additional discussion about the representativeness of the statistical results according to the data we have been able access so far. (Lines 33, 200-201, 206-210, 336-345, 355-356)

In the revised version, we modified our previous conclusion regarding 40°S. A more reasonable range of 30-40°S was proposed, generally covering gauge stations located in Oceania, South America, and Africa in the database. Among all these stations, river syndromes were identified in the Murray-Darling system of Australia (softening syndrome involved at 12 gauge stations), and South Africa (alkalization syndrome at 2 gauge stations).

Again, the authors are very grateful to the anonymous Reviewer for valuable comments and suggestions which are of great significance in improvement of the manuscript quality.

Response to Reviewer #3

This is a really compelling study of longterm changes to solute loads (concentrations and concentrations*discharge) for over 100 major river basins around the world. Such data are challenging to collect in a single place and analyze in a reasonable manner. Clearly the authors have worked diligently to do this here and have a very strong analysis of such data presented here. The novelty of the data, the analyses, and the findings warrant publication, in my opinion. However, it would be helpful if the following concerns were addressed so that the most accurate and appropriate message is communicated to the scientific community (and others).

Response: We thank the reviewer for their insightful suggestions and comments, which we believe have led to a greatly improved manuscript. We have carefully revised the manuscript to address the issues raised by the Reviewer, and to better communicate our message to a broad readership. A point-by-point list of responses is provided as follows.

Moderate Concern:

1. The use of the term 'syndrome' for these sort of long-term changes is novel. I have not seen it used with respect to changes in river solutes before. I believe that the authors are using it appropriately here given their citation of other uses of 'syndrome' in environmental contexts. However, because this is the first use of it for river solutes,

I would advocate for being very careful to articulate exactly what this term means in the context of river solutes. What sort of 'certain thresholds' are needed for these to have been identified for rivers? It seems to me there's a greater analysis of temporal trends here (for each basin) rather than an analysis of specific threshold identification and then exceedance. It would be helpful to resolve this for accuracy of message.

Response: Thank you very much. In the revised manuscript, we have provided a more precise description of the term 'syndrome', from global to river syndromes, and then to syndromes relating to river solutes. We use the term 'solute-induced river syndrome' throughout the revised manuscript. For given trends in river solutes, specific solute-induced syndromes would be expected to appear in the river system once 'certain thresholds' are exceeded.

Statement of the term 'syndrome': A solute-induced river syndrome represents a drastic change to a river system driven by extreme changes in dissolved solids concentration, pH, hardness or alkalinity. This drastic change usually manifests itself as high/low contents of particular solutes, resulting from negative interactions between humans and the environment. Notable solute-induced river syndromes have been observed in the world's large rivers as salinization, mineralization, desalinization, acidification, alkalization, hardening, and softening, all of which are associated with trends in major solutes. Each solute-induced river syndrome, identified according to a certain threshold, is characterized by specific symptoms, impacts, spatial distributions, and representative rivers. Detailed information is provided in revised Table 1. (Lines 185-194)

Thresholds for identification of solute-induced river syndromes: According to Meybeck (2003), the sum of total cation concentrations Σ^+ ($C_{Ca^{2+}} + C_{Mg^{2+}} + C_{Na^+} + C_{K^+}$) < 0.185 meq/L corresponds to ‘extremely dilute’ rivers with least mineralized water, $\Sigma^+ < 0.75$ meq/L to ‘dilute’ rivers, $\Sigma^+ < 1.5$ meq/L to ‘medium dilute’ rivers, 1.5 meq/L $< \Sigma^+ < 3$ meq/L to ‘medium mineralized’ rivers, $\Sigma^+ > 3$ meq/L to ‘mineralized’ rivers, and $\Sigma^+ > 24$ meq/L to ‘saline’ rivers containing the most mineralized water. In combination with trend analysis, we defined the thresholds as follows: salinization with $\Sigma^+ > 24$ meq/L and an increasing trend; mineralization with $\Sigma^+ > 3$ meq/L and an increasing trend; and desalinization with $\Sigma^+ < 1.5$ meq/L and a decreasing trend.

For acidification or alkalization, both pH and alkalinity (or acidity) must be considered. We calculated the ratio of hardness to alkalinity, and extracted mean pH values for rivers from the GEMS database. When the ratio of hardness to alkalinity > 1 , acid input from anthropogenic sources is likely to be responsible (Xia, 1998). We therefore propose the following diagnosis procedure: (1) if the water pH < 7 and the ratio of hardness to alkalinity > 1 and increasing with time in a river, the river is experiencing an acidification syndrome; and (2) if pH > 7 and the ratio of hardness to alkalinity < 1 and decreasing with time, the river is undergoing an alkalization syndrome.

Finally, we examined hardness (expressed as calcium carbonate equivalent in mg/L, $CaCO_3$ mg/L) following the WHO classification of water hardness into soft (< 60 mg/L), moderately hard (60-120 mg/L), hard (120-180 mg/L), and very hard (> 180 mg/L) levels (McGowan, 2000). Therefore, the hardening syndrome is assumed to

occur if river water hardness > 120 mg/L with an increasing trend. Conversely, the softening syndrome is assumed to occur if river water hardness < 60 mg/L with a decreasing trend. (Lines 404-419 in the revised main text, and Lines 190-214 in the revised Supporting Information)

In the revised Fig. 3, we have marked areas where the syndrome appears as given by where the solute metric exceeds certain threshold. In the revised version, we also added Supplementary Fig. 4 to present what sort of 'certain threshold' is needed for a given solute-induced river syndrome under the associated temporal trend in solute concentration.

References

- McGowan W. Water processing: residential, commercial, light-industrial 3rd edn (Water Quality Association, Lisle, 2000).
- Meybeck, M. Global Occurrence of Major Elements in Rivers. *Treatise Geochem.* **5**, 207-223 (2003).
- Xia, X. H. Water quality (major ions) evolution of the Yangtze river system from 1960's to 1990's (Peking University, Beijing, 1998). in Chinese

1 **Revised Table 1** Characteristics of typical solute-induced river syndromes in the world's large rivers

Syndrome (number of sites)	Symptom	Identification criteria	Impact	Continent	Representative river (site)	References
Salinization (2 sites)	Salt contents↑,	$\Sigma^+ (\text{Ca}^{2+} + \text{Mg}^{2+} + \text{Na}^+ + \text{K}^+) > 24$ meq/L, increasing trend	Infrastructure corrosion; Contaminant transport;	/	Rhine River (Maassluis)	38
Mineralization (33 sites)	TDS↑	$\Sigma^+ (\text{Ca}^{2+} + \text{Mg}^{2+} + \text{Na}^+ + \text{K}^+) > 3$ meq/L, increasing trend	Safe drinking water; Biodiversity	Central, Southeast of North America; West of Europe	South Saskatchewan (Medicine Hat)	26, 39
Desalinization (12 sites)	Salt contents↓, TDS↓	$\Sigma^+ (\text{Ca}^{2+} + \text{Mg}^{2+} + \text{Na}^+ + \text{K}^+) < 1.5$ meq/L, decreasing trend	Drinking water supply; River-ocean ecosystem	Oceania	Murray River (Torrumbarry)	40, 41
Acidification (8 sites)	HCO_3^- ↓, pH↓, Hardness↑	pH < 7; Hardness/Alkalinity > 1, increasing trend	Impacts on ecosystem and biodiversity; Infrastructure corrosion	South America; Southeast of North America	Beni River (Rurrenabaque)	42-46
Alkalization (12 sites)	HCO_3^- ↑, pH↑, Hardness↓	pH > 7; Hardness/Alkalinity < 1, decreasing trend	Infrastructure damage, Impacts on irrigation areas, ecosystem and biodiversity	Central North America; Northern Asia; Southern Africa	Groot Vis River (Matomelas Reserve)	47, 48
Hardening (36 sites)	$(\text{Ca}^{2+} + \text{Mg}^{2+})$ ↑, Cation↑	Hardness (CaCO_3 mg/L) > 120, increasing trend	Threat to soils and safe drinking waters; Impacts on ecosystem	North America; Western Europe; South and East of Asia	Yellow River (Toudaoguai)	31
Softening (18 sites)	$(\text{Ca}^{2+} + \text{Mg}^{2+})$ ↓, Cation↓	Hardness (CaCO_3 mg/L) < 60, decreasing trend	Impacts on ecosystem and biodiversity	Central South America; Southeastern Oceania	Roanoke River (Randolph)	49, 50

Revised Fig. 3 River syndromes induced by extreme changes in solutes. A~G represent seven river syndromes associated with ion concentration. The colored area indicates that the river is experiencing a specific syndrome as the solute metric exceeds a certain threshold. The central open circle refers to the ion concentration range when the river ecosystem is in a healthy state. Colored annuli refer to ion concentrations that exceed prescribed thresholds, causing river syndromes. Colored dots in A~G denote measured data, and colored lines indicate the three-year moving average.

Supplementary Fig. 4 Warning signs and thresholds of solute metrics for identification of solute-induced river syndromes. **a** illustrates the thresholds of solute metrics used to identify solute-induced river syndromes. **b~h** present seven river syndromes associated with temporal variation in solute concentration. The critical line indicates the threshold used to identify a solute-induced river syndrome.

The warning line represents the status when a specific solute metric reaches 90% of one of the corresponding thresholds (e.g. Σ^+ ($\text{Ca}^{2+}+\text{Mg}^{2+}+\text{Na}^++\text{K}^+$) at 21.6 meq/L, 2.7 meq/L, and 1.7 meq/L for salinization, mineralization, and desalinization; hardness/alkalinity at 0.9 and 1.1 for acidification and alkalization; and hardness (CaCO_3) at 108 mg/L and 66 mg/L for hardening and softening).

2. Related... One thing I have some trouble figuring out is perhaps related to the use of the term 'syndrome' and the varying changes in solute loads around the world. That is, I typically associate the term 'syndrome' to indicate a single sickness/challenge, etc. that affects many people, but in all cases it can be diagnosed the same way. In this case, the river syndromes are quite varied. In some cases salinization is occurring, in others, freshening (the opposite) is occurring. Further, the authors note that the causes of these different trajectories are somewhat unique across river basins too (i.e., some are from changes in weathering rates, some are from anthropogenic activities). I am not sure how to rectify this except perhaps to (in combination with making edits to reply to #1 above) be explicit that the syndrome is purely about long term changes in solute concentrations and/or loads. Maybe re-name to always be referred to as 'solute syndrome' which allows for any solutes to be included?

Response: Thank you very much for these excellent suggestions. After a careful evaluation of the concept of 'syndrome', we have decided to use the term 'solute-induced river syndromes' throughout the revised version so as to allow any solute to be included. Solute-induced river syndromes are then identified according to

prescribed thresholds of solute metrics, with corresponding trends in the concentrations of solutes indicating when syndromes might occur or how they might develop from either a short-term (based on shorter data length) perspective or a long-term (based on longer data length) perspective. This should be of great significance to researchers and administrators responsible for water quality and river ecosystem management. In the present study, the most notable solute-induced syndromes include salinization, mineralization, desalinization, acidification, alkalization, hardening, and softening, covering the major solutes (Ca^{2+} , Mg^{2+} , Na^+ , K^+ , SO_4^{2-} , Cl^- , HCO_3^-) and dissolved silica. The concept of ‘solute-induced river syndromes’ should allow inclusion of more solutes in the future.

3. The authors recommend attempts to control or curb some activities that may be contributing to syndromes (lines 292-300), but I wonder if they can offer advice about *when* one can determine that a syndrome has occurred or is occurring. Any trend over a few years can be spurious, and yet at the other end of the temporal spectrum, decades of data collection would mean that the syndrome has occurred for too long. How can we detect these events early (and robustly)?

Response: Thank you very much. Similar to the above response to the last question, the solute-induced river syndromes are identified according to prescribed thresholds of solute metrics. The corresponding trends in concentrations of solutes make it possible to estimate when specific syndromes would occur or how they develop from

either a short-term (based on shorter data length) perspective or a long-term (based on longer data length) perspective. As a result, we are able to offer advice concerning the time when a syndrome is occurring or has occurred. In Fig. 3 of the revised manuscript, we highlight rivers that are suffering syndromes due to extreme changes in solute concentrations above certain thresholds. To address the Reviewer's concern, we have added Supplementary Fig. 4 in the revised version to display typical scenarios in terms of the proximity between the existing status of the solute metrics and the prescribed thresholds (see critical lines in the Figure) for the different syndromes of interest. For river management purposes, warning lines have been set for each of the solute-induced river syndromes (Supplementary Fig. 4). The trends in solute concentration suggest river development directions towards or away from the syndromes. The co-varying trends in both runoff and solute concentration potentially provide important information on major causes of river syndromes associated with solutes evolution. (Lines 310-319)

Solute-induced syndromes have been found in several of the world's large rivers. For instance, the Musselshell station of Rhine River (Supplementary Fig. 4b) has exhibited symptoms of salinization since 1985, with the total cation concentration exceeding the 24 meq/L threshold. The increasing trend in total cation concentration implies a deteriorating condition of salinization in following decades, indicating an urgent need for strategies to curtail development of this syndrome. As mentioned above, the co-varying trends in both runoff and solute concentration potentially provide important information on major causes of river syndromes associated with

solutes evolution. At Musselshell, runoff exhibits a stable trend, and so salinization is mainly driven by the solute content itself rather than from reduction in runoff. River syndromes driven by significant changes in both solute concentration and runoff can also be found in other large rivers.

Supplementary Fig. 4 Warning signs and thresholds of solute metrics for

identification of solute-induced river syndromes. **a** illustrates the thresholds of solute metrics used to identify solute-induced river syndromes. **b~h** present seven river syndromes associated with the temporal variation in solute concentration. The critical line indicates the threshold used to identify a solute-induced river syndrome. The warning line represents the status when a specific solute metric reaches 90% of one of the corresponding thresholds (e.g. Σ^+ ($\text{Ca}^{2+}+\text{Mg}^{2+}+\text{Na}^++\text{K}^+$) at 21.6 meq/L, 2.7 meq/L, and 1.7 meq/L for salinization, mineralization, and desalinization; hardness/alkalinity at 0.9 and 1.1 for acidification and alkalization; and hardness (CaCO_3) at 108 mg/L and 66 mg/L for hardening and softening).

It is generally accepted (Knutsson, 1994) that a data length greater than 10 years is preferable for a trend analysis of solute concentration. To detect solute-induced syndromes early and even predict their future evolution, there is an urgent need for global long-term river quality data (Schellnhuber et al., 1997). Due to the present lack of available data, it is recommended that predictive models be developed in conjunction with global long-term monitoring campaigns. Biogeochemical models should be developed that link syndromes to natural conditions and human disturbances of basins, and then applied to predict the future evolution of such syndromes under different scenarios. (Lines 336-345)

References

Knutsson, G. Trends in the acidification of groundwater. *IAHS Publications-Series of Proceedings*

and Reports-International Associations of Hydrological Sciences **220**, 107-120 (1994).

Schellnhuber, H. J., et al. Syndromes of global change. *GAIA-Ecological Perspectives for Science and Society* **6**, 18-33 (1997).

Minor Concern:

1. Perhaps I'm missing this but the emphasis in a few places of the big changes in river basins from 40S to 40N latitude has me thinking that this is also the region of the globe with the greatest human population. I'm not sure that that's true, but it sure seems like it would be. Is there an easy way to figure this out and then perhaps make that correlation (I'm not necessarily advocating for a statistical analysis) here? I see the relationship to urban and agricultural lands in this range of the globe, but ultimately isn't this also about people being particularly densely confined to this region compared to the rest of the globe?

Response: Yes. Population density is an important factor regarding human impacts on solute changes in rivers, particularly in the northern hemisphere. According to Kummu & Varis (2011), about half of the world's population dwell within the area between 20°N and 40°N. Moreover, the most populated latitudes experience considerable water scarcity, owing to their greater proportion of urban and agricultural land use than in other parts of the globe. Li and Bush (2015) reported that HCO_3^- and Mg^{2+} concentrations or yields are correlated with population density in the Mekong basin. Kaushal et al. (2018) indicated that trends in salinization and

alkalization of freshwater are common in humid regions of the eastern United States, where acidic precipitation and human population density are the greatest. The freshwater salinization syndrome is most prominent in the densely populated eastern and midwestern United States, where salinity and alkalinity have increased most rapidly, due to salt pollution (e.g., road deicers, irrigation runoff, sewage, and potash), accelerated weathering and soil cation exchange, mining and resource extraction, and the presence of easily weathered minerals used in agriculture (lime) and urbanization (concrete).

In our study, population is taken as one of the key factors controlling the concentration of solutes, together with other factors such as lithology, climate, land use and land cover, agricultural activities, and dam construction. (Lines 236-238)

For sites with river syndromes, RDA plots related to anthropogenic factors (Fig. 5f and Supplementary Fig. 8b in the revised version) indicated positive correlations between Na^+ , Cl^- and irrigation, agriculture, population, in accordance with findings from the previous study of the Mekong River (Li and Bush, 2015). (Lines 294-298)

Given the significant impacts of solute-induced river syndromes on water quality, contaminant transport, biodiversity, and ecosystem function, we recommend that control of syndromes is prioritized at 'hotspots'. For example, improved management practices are required to prevent severe salinization in central and southeast North America and west Europe where population density is high. (Lines 325-327)

References

- Kummu, M. & Varis, O. The world by latitudes: A global analysis of human population, development level and environment across the north-south axis over the past half century. *Appl. Geogr.* **31**, 495-507 (2011).
- Li, S. Y. & Bush, R. T. Changing fluxes of carbon and other solutes from the Mekong River. *Sci. Rep.* **5**, 1-16 (2015).
- Kaushal, S. S., et al. Freshwater salinization syndrome on a continental scale. *Proc. Natl. Acad. Sci. U. S. A.* **115**, E574-E583 (2018).

Revised Fig. 5 Interpretation of natural and anthropogenic impacts on changes

in solutes with river syndromes. a and b, Gibbs model for stations with solute-induced river syndromes under varying trends in Q . The 60 dots in purple, blue, and yellow respectively represent decreasing, stable, and increasing trends in annual runoff. **c and d** are latitudinal distributions of proportional areas of temperate climate type and urban land area. **e and f** are RDA results for the influence on dissolved solids of natural factors (reclassification for Köppen climate and lithology) and anthropogenic factors (irrigation, population, DOR, and detailed reclassification of

land cover), in critical latitudinal belts.

2. line 133 - not all permafrost is 'ice' or 'ice-rich' so it's better to say that permafrost 'thaws' rather than melts.

Response: Many thanks. We have replaced “permafrost melting” with “permafrost thaw” in the revised manuscript. (Line 145)

Finally, we would like to express our sincere thanks to the anonymous Reviewer for valuable comments and suggestions which are of great importance in improving our manuscript.

REVIEWERS' COMMENTS

Reviewer #1 (Remarks to the Author):

I reviewed a revised version of the manuscript "Global syndromes induced by changes in solutes of the world's large rivers" by Wu et al. I commend the authors on their revised manuscript and have no further comments or edits. In particular I appreciate the authors' efforts to incorporate more of the previous "syndrome" research and a broadening of "hot-belts" across the range of studied latitudes. Additionally, I think the authors have done a nice job emphasizing that many of the observed changes are to the solutes themselves. This points to some very interesting terrestrial/watershed based processes occurring and which are critical for more complete earth-system models.

Reviewer #2 (Remarks to the Author):

I have carefully reviewed the revision of the manuscript and the rebuttal letter. The authors have done an excellent job of thoughtfully responding to both reviews. These results remain noteworthy though the additional context, clarification, and other edits have made this a stronger paper.

I have only 1 minor suggested edit. I see the notation of using the LOADEST model in the methods section. I suggest citing the model in a formal way (perhaps the documentation for the model).

I look forward to this being published and becoming an impactful communication of long-term changes to water quality around the world.